# Genome-wide analysis yields new loci associating with aortic valve stenosis

Anna Helgadottir ID

Aortic valve stenosis (AS) is the most common valvular heart disease, and valve replacement is the only definitive treatment. Here we report a large genome-wide association (GWA) study of 2,457 Icelandic AS cases and 349,342 controls with a follow-up in up to 4,850 cases and 451,731 controls of European ancestry. We identify two new AS loci, on chromosome 1p21 near *PALMD* (rs7543130; odds ratio (OR) = 1.20, $P = 1.2 \times 10^{-22}$) and on chromosome 2q22 in *TEX41* (rs1830321; OR = 1.15, $P = 1.8 \times 10^{-13}$). Rs7543130 also associates with bicuspid aortic valve (BAV) (OR = 1.28, $P = 6.6 \times 10^{-10}$) and aortic root diameter ($P = 1.30 \times 10^{-8}$), and rs1830321 associates with BAV (OR = 1.12, $P = 5.3 \times 10^{-3}$) and coronary artery disease (OR = 1.05, $P = 9.3 \times 10^{-5}$). The results implicate both cardiac developmental abnormalities and atherosclerosis-like processes in the pathogenesis of AS. We show that several pathways are shared by CAD and AS. Causal analysis suggests that the shared risk factors of Lp(a) and non-high-density lipoprotein cholesterol contribute substantially to the frequent co-occurence of these diseases.

#A full list of authors and their affliations appears at the end of the paper.

Aortic valve stenosis (AS) is characterized by thickened and calcified valvular cusps causing left ventricular outflow obstruction. This progressive disease is usually graded as mild, moderate, or severe, based on the valve area and pressure gradient across the valve. Severe AS is a notable cause of morbidity and mortality, affecting approximately 5% of those over 70 years of age[1–3], and the estimated 5-year survival in symptomatic severe AS ranges from 15 to 50% unless outflow obstruction is relieved by aortic valve replacement[3].

The pathogenesis of the disease remains poorly understood. However, several of the associated clinical risk factors of calcified aortic valve are shared by atherosclerotic disease, and immunohistochemical studies show that calcified aortic valve lesions have many characteristic features of atherosclerosis, including initial endothelial damage, oxidized lipid deposition, chronic inflammation, and calcification[4]. In addition, bicuspid aortic valve (BAV), the most common congenital cardiac malformation, when the aortic valve has two leaflets instead of three, accelerates the development of AS by decades[4]. While the prevalence of BAV is 0.5–2% in the population, BAV is found in up to half of those with severe AS[5].

Little is known about the genetics of AS, although a recent genome-wide association (GWA) study reported the association of rs10455872 in the *LPA* gene, encoding apolipoprotein(a) of lipoprotein (a) (Lp(a)), with calcification of the aortic valve, and with AS[6]. Elevated serum levels of Lp(a) have also been associated with increased risk of AS[7]. These findings are in keeping with a common pathogenic feature of AS and atherosclerosis[8,9].

Another genetic study recently showed that a rare p.Arg721Trp *MYH6* missense variant, which was previously shown to associate with sick sinus syndrome and atrial fibrillation[10,11], also associates with coarctation of the aorta, BAV, and with AS[12].

Here, we describe a large GWA study of AS including 2,457 cases and 349,342 controls, with follow-up in up to 4,850 AS cases and 451,731 controls. We examined the association of AS variants with BAV and several other cardiovascular conditions and assessed the shared genetic risk of AS and coronary artery disease (CAD).

## Results

**Novel variants associate with aortic stenosis.** We tested 32.5 million sequence variants for association with AS in 2,457 Icelandic cases and 349,342 controls (see Manhattan plot in Supplementary Fig. 1). We identified the variants by whole-genome sequencing 15,220 Icelanders, and imputed them into 151,678 chip-typed, long-range phased individuals and their close relatives[13].

We observed one genome-wide significant association, between AS and the intergenic variant rs7543130 (effect allele frequency (EAF) [A] = 51.2%) on chromosome 1p21 near the *PALMD* gene (odds ratio (OR) = 1.23; 95% confidence interval (CI): 1.15–1.31, $P = 6.8 \times 10^{-10}$ (significance threshold for intergenic variants set at $P = 7.9 \times 10^{-10}$, see Methods and ref.[14])) (Table 1). We noted that rs7543130 was recently reported to associate with aortic root size[15] and we replicate this association in our Icelandic aortic root dimension sample ($P = 1.3 \times 10^{-8}$) (Table 2).

We tested the top seven common and low-frequency variants in the discovery GWA scan, including rs7543130, in up to 4,850 AS cases and 451,731 controls from Sweden, Norway, United Kingdom, and the United States (Table 1, Supplementary Data 1). The joint analysis showed a robust association between AS and rs7543130 (OR = 1.20; 95% CI: 1.16–1.25; $P = 1.2 \times 10^{-22}$) as well as rs1830321 (EAF[T] = 37.5%) intronic to *TEX41*, a non-protein coding gene on chromosome 2q22 (OR = 1.15; 95% CI: 1.11–1.20, $P = 1.8 \times 10^{-13}$) (Table 1).

We replicated the reported association of the intronic *LPA* variant[6] rs10455872 with AS in Iceland and the follow-up sample sets (combined OR = 1.46; 95% CI: 1.37–1.56, $P = 1.9 \times 10^{-31}$) (Table 1). In contrast, we did not find association with variants implicating osteogenic and calcium signaling pathway genes, previously reported to suggestively associate with AS[16] ($P > 0.05$ in Iceland and UK Biobank).

We tested the association of the two novel AS variants and the *LPA* variant with a subset of Icelandic AS cases who had undergone aortic valve replacement, representing those with severe AS. Although less significant, likely due to smaller sample size, the effect sizes were not significantly different from those for all AS (Supplementary Data 2).

**Aortic stenosis variants and cardiovascular phenotypes.** We tested the rs7543130 near *PALMD*, rs1830321 in *TEX41*, and the *LPA* rs10455872, for association with BAV, a major risk factor for AS[4,5], in 1,555 cases and 33,883 controls from Iceland, Sweden, and the United States. Both of the novel AS variants associate with BAV and the rs7543130 association was genome-wide significant (OR = 1.28; 95% CI: 1.19–1.39; $P = 6.6 \times 10^{-10}$; OR = 1.12, 95% CI: 1.04–1.22, $P = 5.3 \times 10^{-3}$ for rs1830321). The *LPA* rs10455872 does not associate with BAV (Table 2).

Table 2 also shows the association of the rare p.Arg721Trp *MYH6* missense variant rs387906656 (EAF = 0.34%) with the risk of BAV (OR = 8.04; 95% CI: 3.36–19.22; $P = 2.8 \times 10^{-6}$). This variant was previously shown to associate with sick sinus syndrome and atrial fibrillation[10,11], and was recently reported also to associate with coarctation of the aorta, BAV, and AS

---

### Table 1 Meta-analysis results for aortic valve stenosis variants

| | Cases/controls | *PALMD* intergenic rs7543130 [A/C] EAF = 51.2% | | *TEX41* intronic rs1830321 [T/C] EAF = 37.5% | | *LPA* intronic rs10455872 [G/A] EAF = 6.2% | |
|---|---|---|---|---|---|---|---|
| | | OR (95% CI) | *P* value | OR (95% CI) | *P* value | OR (95% CI) | *P* value |
| Iceland | 2457/349,342 | 1.23 (1.15–1.31) | $6.8 \times 10^{-10}$ | 1.20 (1.12–1.28) | $7.6 \times 10^{-8}$ | 1.4 (1.23–1.56) | $1.8 \times 10^{-7}$ |
| Sweden (MDCS)[a] | 470/15,162 | 1.14 (0.98–1.33) | 0.092 | 1.21 (1.05–1.39) | 0.0080 | 1.55 (1.28–1.88) | $1.0 \times 10^{-5}$ |
| Sweden, Stockholm | 318/1376 | 1.25 (0.98–1.59) | 0.068 | 1.18 (0.89–1.56) | 0.24 | 1.19 (0.90–1.57) | 0.23 |
| UK Biobank | 1844/406,814 | 1.25 (1.17–1.33) | $3.8 \times 10^{-11}$ | 1.14 (1.06–1.22) | $1.36 \times 10^{-4}$ | 1.54 (1.38–1.71) | $4.8 \times 10^{-15}$ |
| Norway (HUNT) | 1546/24,235 | 1.13 (1.05–1.22) | 0.0012 | 1.11 (1.02–1.20) | 0.010 | 1.48 (1.28–1.71) | $1.0 \times 10^{-7}$ |
| USA, Michigan | 251/2510 | 1.15 (0.96–1.39) | 0.13 | 1.01 (0.85–1.24) | 0.76 | 1.32 (0.94–1.84) | 0.10 |
| Combined | 6886/799,439 | 1.20 (1.16–1.25) | $1.2 \times 10^{-22}$ | 1.15 (1.11–1.20) | $1.8 \times 10^{-13}$ | 1.46 (1.37–1.56) | $1.9 \times 10^{-31}$ |

Results are shown for the discovery and follow-up datasets and the joint analysis (combined). The effect allele is the first allele in brackets [effect allele/non-effect allele]. The EAF is for the Icelandic population. *P* value from logistic regression analysis. Results from the different study groups were combined using a Mantel–Haenszel model
*EAF* effect allele frequency, *OR* allelic odds ratio, 95% *CI* 95% confidence interval, *MDCS* Malmö Diet and Cancer study
[a] The association results for the rs10455872 variant in the MDCS included 613 cases and 28,109 controls

**Table 2 Association of aortic valve stenosis variants with other cardiovascular traits**

| Bicuspid aortic valve | 208/25,139 | PALMD intergenic rs7543130 [A/C] | | TEX41 intronic rs1830321 [T/C] | | LPA intronic rs10455872 [G/A] | | MYH6 missense rs387906656 [A/G] p.Arg721Trp | |
|---|---|---|---|---|---|---|---|---|---|
| | | OR (95% CI) | P value | OR (95% CI) | P value | OR (95% CI) | P value | OR (95% CI) | P value |
| Bicuspid aortic valve | 208/25,139 | 1.26 (0.99, 1.60) | 0.059 | 1.31 (1.03, 1.67) | 0.025 | 1.12 (0.72, 1.75) | 0.61 | 8.04 (3.36, 19.22) | $2.8 \times 10^{-6}$ |
| Sweden-Stockholm | 275/1516 | 1.27 (1.06, 1.52) | 0.0098 | 1.20 (0.99, 1.46) | 0.063 | 1.02 (0.92, 1.12) | 0.77 | | |
| USA-Houston | 147/864 | 1.29 (0.99, 1.67) | 0.057 | 1.25 (0.97, 1.60) | 0.085 | 0.96 (0.51, 1.81) | 0.91 | | |
| USA-Boston | 452/1834 | 1.27 (1.09, 1.48) | 0.002 | 1.10 (0.95, 1.28) | 0.21 | 1.44 (1.08, 1.93) | 0.014 | | |
| USA-Michigan | 473/4730 | 1.31 (1.14, 1.51) | $1.2 \times 10^{-4}$ | 1.00 (0.86, 1.16) | 0.97 | 1.19 (0.93, 1.54) | 0.17 | | |
| Combined BAV | 1555/33,883 | 1.28 (1.19, 1.39) | $6.6 \times 10^{-10}$ | 1.12 (1.04, 1.22) | $5.3 \times 10^{-3}$ | 1.07 (0.98, 1.16) | 0.13 | | |
| Atrial septal defect | 708/353,019 | 1.23 (1.07, 1.42) | $3.9 \times 10^{-3}$ | 1.22 (1.06, 1.41) | $5.9 \times 10^{-3}$ | 1.15 (0.87, 1.52) | 0.32 | 3.17 (1.47, 6.81) | $3.2 \times 10^{-3}$ |
| Ventricular septal defect | 902/357,428 | 1.23 (1.07, 1.42) | $4.8 \times 10^{-3}$ | 1.04 (0.90, 1.21) | 0.59 | 1.14 (0.84, 1.53) | 0.41 | 4.40 (2.14, 9.07) | $5.7 \times 10^{-5}$ |
| Coronary artery disease | 37,782/318,845 | 1.00 (0.97, 1.02) | 0.74 | 1.05 (1.03, 1.08) | $9.3 \times 10^{-5}$ | 1.28 (1.21, 1.34) | $2.4 \times 10^{-22}$ | 1.21 (1.00, 1.48) | 0.056 |
| Phenotype (qtl) | N | β (SE) | P value | β (SE) | P value | β (SE) | P value | β (SE) | P value |
| Aortic root diameter | 19,513 | 0.065 (0.01) | $1.3 \times 10^{-8}$ | −0.017 (0.02) | 0.16 | −0.052 (0.02) | 0.020 | −0.068 (0.08) | 0.40 |

Association of aortic valve stenosis variants with cardiovascular phenotypes is shown for Icelandic samples. Follow-up and joint analysis (combined BAV) is also provided for the association with BAV. The effect allele is the first allele in brackets [effect allele/non-effect allele]. The effect (β) for aortic root diameter is given in standardized units. Logistic (cc) or linear (qtl) regression analyses were used for association testing. Results from the different study groups were combined using a Mantel–Haenszel model
Cc case−control, Qtl quantitative trait, OR allelic odds ratio, 95% CI 95% confidence interval, BAV bicuspid aortic valve, SE standard error

**Table 3 Coronary artery disease variants and aortic root size variants that associate with aortic valve stenosis**

| Primary association | Locus | Chr. | Coding effect | Rs name | EA/other allele | OR (95% CI) | P value | $P_{het}$ | $I^2$ |
|---|---|---|---|---|---|---|---|---|---|
| CAD | CELSR2/PSRC1 | 1 | Downstream gene | Rs646776 | T/C | 1.11 (1.05-1.18) | $3.4 \times 10^{-4}$ | 0.82 | 0 |
| CAD | LPA | 6 | Missense (p.Ile1891Met) | Rs3798220 | C/T | 1.55 (1.33-1.81) | $2.1 \times 10^{-8}$ | 0.4 | 0 |
| CAD | SH2B3 | 12 | Missense (p.Trp60Arg) | Rs3184504 | C/T | 0.91 (0.87-0.96) | $1.6 \times 10^{-4}$ | 0.94 | 0 |
| Aortic root size | CFDP1[a] | 16 | Intronic | Rs17696696 | G/T | 1.07 (1.03-1.11) | $1.3 \times 10^{-4}$ | 0.055 | 60.5 |
| CAD | ANGPTL4[b] | 19 | Missense (p.Glu40Lys) | Rs116843064 | A/G | 0.77 (0.68-0.88) | $9.5 \times 10^{-5}$ | 0.36 | 8.6 |

Shown are CAD variants and aortic root size variants that associate with AS. A total of 71 CAD and 11 aortic root size variants from genome-wide association studies were tested (primary association). The CELSR2/PSRC1, LPA, and SH2B3 variants were tested in 4,301 AS cases and 756,156 controls from Iceland and the UK Biobank. Results from the different study groups were combined using a Mantel–Haenszel model
P values for the combined analyses are provided
CAD coronary artery disease, AS aortic valve stenosis, Chr.: chromosome, EA effect allele, OR odds ratio, $P_{het}$: P value for heterogeneity between study groups, $I^2$: heterogeneity $I^2$ statistics for the combined analysis
[a] The CFDP1 variant was tested in 6,416 cases and 784,277 controls (additional samples from Sweden-Stockholm, Norway-HUNT, and the USA, Michigan)
[b] The ANGPTL4 variant was tested in 6,886 cases and 799,439 controls (same samples as for CFDP1 plus samples from Sweden-MDCS)

(OR = 2.65; 95% CI: 1.78–3.96; $P = 1.8 \times 10^{-6}$)[12]. The effect size on BAV is substantially greater than that for AS ($P = 0.023$), suggesting that the AS risk conferred by this variant is mediated through BAV.

Next, we examined the association of the AS variants with several other cardiovascular diseases in Icelandic data. In line with the BAV association of p.Arg721Trp in MYH6, rs7543130, and rs1830321, all three variants associate with ventricular defects and/or atrial septal defects ($P < 0.006$) (Table 2 and Supplementary Data 2).

Like the LPA variant, rs1830321 in TEX41 associates with CAD in Iceland (OR = 1.05, 95% CI:1.03–1.08; $P = 9.3 \times 10^{-5}$), but the MYH6 missense variant and rs7543130 near PALMD do not (Table 2 and Supplementary Data 2). The TEX41 rs1830321 is in linkage disequilibrium (LD) with a known GWA CAD variant rs2252641 at the same locus ($R^2 = 0.80$)[17].

Given that several atherosclerosis risk factors have been associated with AS[6,18,19], we tested the novel AS variants for association with the traditional cardiovascular risk factors and observed a nominally significant association ($P < 0.02$) between rs1830321 and systolic and diastolic blood pressure in Iceland (Supplementary Data 3) and in data from the UK Biobank (https://biobankengine.stanford.edu/search#).

**Shared genetic risk factors with CAD.** The frequent comorbidity of CAD and AS[20], together with the similarities in histopathology[4], suggest shared genetic predisposition. Therefore, we tested 71 CAD variants[21,22] for association with AS, both individually (Supplementary Data 4 and Table 3) and as a weighted genetic risk score (CAD-GRS-all) (Table 4). We excluded from this analysis the LPA variant rs10455872 and rs1830321 in TEX41 that associate genome-wide significantly with both CAD and AS.

In the Icelandic and UK Biobank datasets combined, four CAD variants associate with AS at a significance threshold set at $P = 7.0 \times 10^{-4} = 0.05/71$. These are the LPA variant rs3798220 (p.Ile1891Met), rs116843064 in ANGPTL4 (p.Glu40Lys), rs646776 at the CELSR2/PSRC1 locus, and rs3184504 in SH2B3 (p.Trp60Arg) (Table 3 and Supplementary Data 4 and 5).

Consistent with a shared genetic risk, the CAD-GRS-all associates with AS both in the Icelandic and the UK Biobank datasets (combined $P = 7.5 \times 10^{-9}$) (Table 4). However, the effect on AS is only 37% of the effect on CAD and the AS association is not significant after adjustment for CAD diagnosis. Given the reported association between genetic predisposition to both elevated Lp(a) and low-density lipoprotein (LDL) cholesterol and AS[6,19], we tested a subset of CAD-GRS (labeled as CAD-GRS-lip), constructed based on 14 Lp(a) and LDL cholesterol/non-high-density lipoprotein (HDL) cholesterol variants (Supplementary Data 4) for association with AS. The CAD-GRS-lip associated strongly with AS ($P = 5.1 \times 10^{-19}$) with an effect that was similar or larger than that for CAD ($β = 0.91$ and 1.02, for CAD and AS, respectively). This association with AS remained after adjusting for CAD diagnosis ($P = 5.1 \times 10^{-9}$) (Table 4), suggesting that the genetic predisposition to elevated Lp(a) and LDL cholesterol explains in large part the shared genetic risk between CAD and AS. Specifically, examining the impact of CAD-GRS-lip on the risk of AS among CAD cases shows that CAD cases with genetic predisposition to high Lp(a) or LDL/non-HDL cholesterol are at a greater risk of having AS than CAD

**Table 4 The association of coronary artery disease genetic risk score with aortic valve stenosis**

| | | Iceland | | UK Biobank | | Iceland + UK Biobank combined | |
|---|---|---|---|---|---|---|---|
| | | $\beta$ | P value | $\beta$ | P value | $\beta$ (95% CI) | P value |
| CAD-GRS-*all* | CAD | 0.77 | $5.2 \times 10^{-153}$ | 0.76 | $<10^{-300}$ | 0.76 (0.73–0.79) | $<10^{-300}$ |
| . | AS | 0.29 | 0.00027 | 0.28 | $7.2 \times 10^{-6}$ | 0.28 (0.19–0.38) | $7.5 \times 10^{-9}$ |
| . | $AS_{adj.CAD}$ | 0.03 | 0.62 | −0.05 | 0.45 | −0.01 (−0.10, 0.08) | 0.83 |
| CAD-GRS-*lip* | CAD | 0.89 | $1.4 \times 10^{-36}$ | 0.92 | $7.8 \times 10^{-99}$ | 0.91 (0.84–0.99) | $1.4 \times 10^{-134}$ |
| . | AS | 1.05 | $2.3 \times 10^{-9}$ | 0.99 | $3.8 \times 10^{-11}$ | 1.02 (0.79–1.24) | $5.1 \times 10^{-19}$ |
| . | $AS_{adj.CAD}$ | 0.77 | $1.5 \times 10^{-5}$ | 0.60 | $6.6 \times 10^{-5}$ | 0.67 (0.45–0.90) | $5.1 \times 10^{-9}$ |
| CAD-GRS-*non-lip* | CAD | 0.73 | $6.3 \times 10^{-120}$ | 0.72 | $4.3 \times 10^{-293}$ | 0.73 (0.69–0.78) | $<10^{-300}$ |
| . | AS | 0.14 | 0.074 | 0.14 | 0.048 | 0.14 (0.04–0.24) | 0.0076 |
| . | $AS_{adj.CAD}$ | −0.11 | 0.15 | −0.18 | 0.0088 | −0.15 (−0.25, −0.05) | 0.0036 |

CAD-GRS-*all* (based on 71 reported CAD variants). CAD-GRS-*lip* (based on 14 CAD variants with reported association with LDL cholesterol (or non HDLcholesterol), or variants at the *LPA* locus. GRS-*non-lip* (based on 57 CAD variants, the same as in CAD-GRS-*all*, but excluding variants in CAD-GRS-*lip*). The effects of the GRSs on CAD are shown for comparison. Logistic regression was used for association testing. Results from the different study groups were combined using a Mantel–Haenszel model
Number of cases–controls in Iceland and UK Biobank, respectively: CAD = 17,488/124,620 and 26,384/382,294; AS = 1,591/140,517 and 1,844/406,814
P value represented as $<10^{-300}$ is $<1 \times 10^{-300}$
GRS genetic risk score, CAD coronary artery disease, AS aortic valve stenosis

cases without such predisposition ($P = 8.5 \times 10^{-7}$) (Supplementary Data 6). In contrast, the complementary subset of CAD-GRS-all (CAD-GRS-non-lip), in which the Lp(a) and LDL/non-HDL cholesterol variants are excluded, associates with less risk of AS after adjusting for CAD status ($P = 0.0036$) (Table 4).

**Aortic root size variants and aortic stenosis.** Given the known association of rs7543130[A] on chromosome 1p21 with increased aortic root dimension[15] and the recognized relationship between BAV and aortopathy[5,23], we excluded known or suspected BAV cases from our echocardiogram database and re-examined the association of this variant with aortic root size. The association with aortic root size remained in this data ($\beta = 0.062$, $P = 4.4 \times 10^{-8}$) (Supplementary Data 2).

We then tested 11 other reported aortic root size variants[15] for association with AS in Icelandic and UK Biobank datasets (Supplementary Data 7). One of these variants, rs17696696[G] intronic to *CFDP1*, associates with AS in these samples and was thus tested in additional 2,115 AS cases and 28,121 controls; the joint analysis yielded OR = 1.07, 95% CI: 1.03–1.11, $P = 0.00013$ (Table 3). A correlated variant rs4888378 ($R^2 = 0.98$) has been reported to associate with carotid intima–media thickness and with the risk of CAD[24]. We also observed a previously unreported genome-wide significant association with CAD in Iceland and the UK Biobank data (combined OR for rs17696696[G] = 1.05, 95% CI: 1.03–1.07, $P = 1.4 \times 10^{-10}$). The AS and CAD risk allele of rs17696696[G] in *CFDP1* associates with smaller aortic root diameter. None of the other AS variants associated with aortic root size (Table 2).

**Candidate causal variants and genes.** Attempting to identify candidate causal variants and genes at the *PALMD* and *TEX41* loci, we first looked for association of the AS variants with expression quantitative trait loci (eQTL) using the Genotype-Tissue Expression dataset[25]. Assessment of 44 diverse human tissues from adults indicated association of rs1830321 with *TEX41* expression, albeit limited to thyroid tissue, but no eQTLs were observed for rs7543130.

To further investigate potential functional relevance of the two AS variants, we mapped variants in LD ($R^2 > 0.5$) with rs7543130 near *PALMD* and rs1830321 in *TEX41* to regulatory regions in heart and aorta tissue samples using public data from the NIH Roadmap Epigenomics Consortium[26,27]. Subsequently, we used chromatin interaction maps[28] for aorta and left and right ventricular heart tissue samples to look for interactions between

the regulatory regions, to which AS risk variants mapped, and gene promoters.

At the *PALMD* locus, four variants (rs11166276, rs6702619, rs1890753, and rs2392040) mapped to three distinct regulatory regions annotated as enhancers and poised promoter (Fig. 1a, upper panel). Multiple chromatin interactions were observed for the regulatory regions harboring these four variants in left ventricular samples. Notably, only the region harboring rs1890753 ($R^2 = 0.97$ with rs7543130) interacted with promoters of genes (Fig. 1a, lower panel). Thus, rs1890753 represents the candidate causal variant at the chromosome 1p21 locus, by directly interacting with the promoters of *PALMD*, *SNX7*, *PLPPR5*, and *PLPPR4*, and the non-coding RNAs, *LOC100129620* and *LOC101928270*. In fetal heart tissue, a poised promoter state is found at the rs1890753 locus (Fig. 1a, upper panel).

At the *TEX41* locus, five variants in LD with rs1830321 overlapped with four distinct regulatory regions (Fig. 1b, upper panel). Chromatin interaction mapping in left ventricular tissue identified the regulatory regions harboring all five variants (rs13028626, rs6749506, rs2252654, rs4662414, and rs13408842) in direct contact to the promoter region of *ZEB2*, and the non-coding RNAs *ZEB2-AS1* and *LINC01412* (Fig. 1b, lower panel). In addition, the rs13408842 region directly interacted with the promoter of *GTDC1* and the non-coding RNA genes *TEX41* and *LOC101928386*. Pairwise correlations ($R^2$) between the five variants and the lead variant rs1830321 ranged from 1.0 for rs13028626, to 0.61 for rs2252654.

Chromatin interactions between the regulatory regions harboring candidate causal variants at the *PALMD* and *TEX41* loci were much less frequent in right ventricular tissue and aorta, compared with the left ventricle, and none overlapped with gene promoters (Supplementary Fig. 2).

## Discussion

Through a large GWA study, we have discovered two common AS variants on chromosomes 1p21 near *PALMD* and 2q22 in *TEX41*, and replicated the previously reported AS variant in *LPA*[6]. Like the rare AS variant in *MYH6*[12], both of the novel AS variants also associate with BAV and congenital cardiac septal defects. The chromosome 2q22 variant also associates with CAD risk.

Given that BAV is a major risk factor for AS, and that the *MYH6* and chromosome 1p21 variants have substantially greater effects on BAV than on AS, it may be postulated that the AS risk conferred by these variants is mediated through BAV. However,

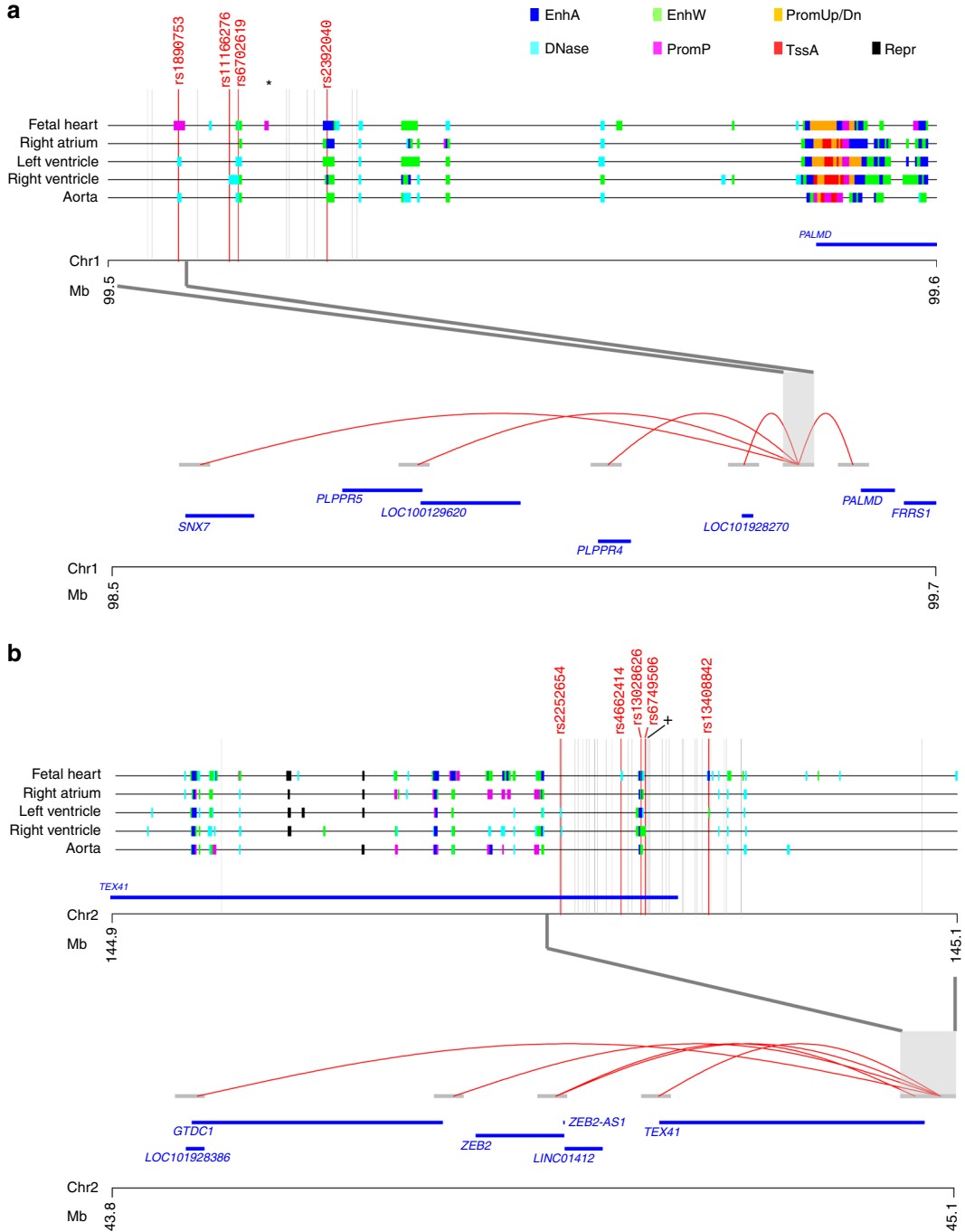

**Fig. 1** Chromatin interactions between regulatory regions harboring candidate causal variants at the *PALMD* and *TEX41* loci. Chromatin states indicative of regulatory regions for the aortic valve stenosis locus on chromosomes 1p21 (**a**) and 2q22 (**b**) are shown for heart and aorta tissue samples. Different types of regulatory states are indicated with distinct colors shown at the top of the figure. EnhA (Enhancer Active), EnhW (Enhancer Weak), PromUp/Dn (Chromatin marks characteristic of a promoter region found upstream or downstream of TSS), DNase (DNase, nucleosome-free/open chromatin region), PromP (Promoter poised region, marked simultaneously as active and repressed, poised for activation during development), TssA (Transcription Start Site, Activated), and Repr (Repressive marks, heterochromatin). Vertical gray lines indicate the variants found in LD ($R^2 > 0.50$) with **a** rs7543130 (*) ($N = 19$) or **b** rs1830321 (+) ($N = 50$). Variants found to overlap with regulatory regions in any of the five tissues are marked up and indicated as red vertical lines. Long-range chromatin interactions in left ventricle tissue samples are shown for **a** the region harboring rs1890753 on chromosome 1p21 with red curved lines, including interactions to promoters for *PALMD*, *PLPPR4*, *PLPPR5*, *DPH5* and *SNX7*, *LOC100129620* and *LOC101928270*, and for **b** regions harboring rs13028626, rs6749506, rs2252654, rs4662414, and rs13408842 that directly interact with the promoter regions of *ZEB2*, *GTDC1*, *ZEB2-AS1*, *LINC01412* and *TEX41*

as we have limited information on whether AS occured on the background of bicuspid or tricuspid valve, we were not able to determine whether these variants associate with AS in the absence of BAV.

Interestingly, the AS and BAV risk allele of rs7543130 near *PALMD* also associates with increased aortic root size. The relationship between BAV and aortopathy is well recognized and several studies suggest that the dilation of the proximal ascending aorta results from changes in flow secondary to the presence of BAV[5,23,29]. This raises the question whether the effect of chromosome 1p21 variant on aortic root size can be explained by its association with BAV. However, our results indicate that the variant's impact on aortic root size is not merely a consequence of BAV, since excluding BAV cases from the analysis had minimal effect on the association. We also note that the *MYH6* missense variant has a large effect on BAV but no effect on aortic root size. We did not find a consistent relationship between genetic associations with risk of AS and aortic root size, but found that one additional aortic root size variant, rs17696696 intronic to *CFDP1*, associates with AS.

We demonstrate that this new AS variant, rs17696696 in *CFDP1*, associates genome-wide significantly with CAD, like rs1830321 near *TEX41* and the *LPA* rs10455872 (ref. [6]). Further, we found that four other CAD variants associate with AS, supporting the notion that there may be a cause shared by CAD and AS. However, contesting a generalized common pathophysiology, causal analysis suggests that only some genetic pathways are shared by CAD and AS, and that the risk of both diseases conferred by Lp(a) and LDL/non-HDL cholesterol levels contributes substantially to the frequent co-occurence of these two diseases.

These results support the assumption that lowering Lp(a) and non-HDL cholesterol levels might slow or prevent progression of AS. However, in a randomized trial of 1,873 patients with mild-to-moderate AS who received statin plus ezetimibe therapy or placebo, active therapy did not reduce the composite outcome of combined aortic valve events and ischemic events during a median follow-up of 52.2 months[30]. It remains conceivable that non-HDL cholesterol lowering therapy implemented earlier in the disease process could affect development of AS. Other potential therapeutic interventions such as Lp(a) lowering among those with high Lp(a) levels need to be explored.

Although we cannot establish how the AS variants at *PALMD* and *TEX41* affect the pathogenesis of disease, chromatin conformational experiments provide clues about potential mechanisms. These experiments show folding of chromatin such that distinct regulatory regions harboring variants in high LD with the lead AS variants, physically interact with several gene promoters, suggesting several candidate causal genes at both loci. Interestingly, in line with an impact during fetal development, a poised promoter state was found in fetal heart tissue for a candidate causal variant rs1890753 at chromosome 1p21. Poised promoters are considered to be involved in the expression of developmental genes allowing for a rapid response to differentiation signals.

At the *TEX41* locus, we note that *ZEB2*, one of the genes suggested through chromatin interaction studies, is a strong biological candidate. ZEB2 is a DNA-binding transcriptional repressor that interacts with activated SMADs, the transducers of tumor growth factor-β (TGFβ) signaling. TGFβs are known to play a role in cardiac development and in several aspects of cardiovascular physiology ranging from the effect on cardiomyocyte and vascular smooth muscle, and renal control of blood pressure[31].

In summary, we discovered two AS variants on chromosomes 1p21 and 2q22. Associations of these and two previously reported AS variants with BAV, other congenital heart defects, aortic root size, and CAD, involve both cardiac developmental abnormalities

and atherogenesis-like processes in the pathogenesis of AS. Further, we demonstrate that four CAD variants and one aortic root size variant associate with AS. While our genetic causal analysis does not support a generalized sharing of genetic risk between CAD and AS, it indicates that the shared risk factors of Lp(a) and non-HDL cholesterol contribute substantially to the frequent co-occurence of these diseases.

## Methods

**deCODE discovery study subjects.** The Icelandic AS sample set included all patients diagnosed in the years 1983–2016 with AS at Landspitali – The National University Hospital (LUH) in Reykjavik, the only tertiary referral center in Iceland. Case status was assigned based on ICD-10 codes I35.0 or I35.2 for discharge diagnoses, or the relevant NOMESCO classification of surgical procedure codes FMA, FMSA, FMD or FMSD, and subcodes). A total of 2,609 cases were identified and of those 2,457 had available genotypes and were included in the analysis. The controls included 349,342 population controls from the Icelandic genealogical database and individuals recruited through different genetic studies at deCODE genetics. The study was approved by the Icelandic Data Protection Authority and the National Bioethics Committee of Iceland (approval no. VSNb2015030022/03.01 with amendments). All participating subjects donating biological samples signed informed consents. Personal identities of the participants and biological samples were encrypted by a third-party system approved and monitored by the Icelandic Data Protection Authority.

The deCODE genetics phenotype database contains extensive medical information on various diseases and traits. Cardiovascular phenotypes used for the purpose of the study included CAD ($N = 37,782$)[32], heart failure ($N = 10,480$), ischemic stroke ($N = 8,948$)[33], atrial fibrillation ($N = 13,471$)[34], sick sinus syndrome ($N = 3,310$)[10], high-degree atrioventricular block ($N = 1,303$), BAV ($N = 208$), atrial septal defect ($N = 708$), ventricular septal defect ($N = 902$), coarctation of aorta ($N = 119$), thoracic aortic aneurysm (TAA) and dissection ($N = 500$), hypertension ($N = 54,974$), type 2 diabetes ($N = 11,448$)[35], and aortic root diameter ($N = 19,506$). The heart failure, high-degree atrioventricular block, coarctation of aorta, TAA and dissection, atrial septal defect, and ventricular septal defect sample sets were based on discharge diagnoses from LUH. Hypertension diagnoses were obtained from the Primary Health Care Clinics of the Reykjavik area, or from LUH. The BAV sample set included individuals with a documentation of BAV in an echocardiographic report from LUH between 1994 and 2015. Measurements of aortic root diameter were obtained from a database of 53,122 echocardiograms from 27,460 individuals performed at LUH between 1994 and 2015. Non-HDL cholesterol measurements ($N = 136,326$) were obtained from three of the largest clinical laboratories in Iceland: (i) LUH (hospitalized and ambulatory patients); (ii) the Laboratory in Mjódd, Reykjavík (ambulatory patients); and (iii) Akureyri Hospital, Regional Hospital in North Iceland, Akureyri (hospitalized and ambulatory patients)[32]. Blood pressure measurements ($N = 125,647$) were obtained from the Primary Health Care Clinics of the Reykjavik area. Measurements were adjusted for sex, year of birth, and age at measurement, and were subsequently standardized to have a normal distribution.

**Whole-genome sequencing and imputation.** This study is based on whole-genome sequence data from 15,220 Icelanders participating in various disease projects at deCODE genetics. In addition, 151,677 Icelanders have been genotyped using Illumina SNP chips and genotype probabilities for untyped relatives are calculated based on Icelandic genealogy. The sequencing was done using Illumina standard TruSeq methodology to a mean depth of 35× (SD 8)[13]. Autosomal single-nucleotide polymorphisms (SNPs) and INDEL's were identified using the Genome Analysis Toolkit version 3.4.0[36]. Information about haplotype sharing was used to improve variant genotyping, taking advantage of the fact that all sequenced individuals had also been chip-typed and long-range-phased[37].

**Genotype imputation information.** The informativeness of genotype imputation (imputation information) was estimated by the ratio of the variance of imputed expected allele counts and the variance of the actual allele counts:

$$\frac{\mathrm{Var}(E(\theta|\text{chip data}))}{\mathrm{Var}(\theta)},$$

where $\theta$ is the allele count. Here, $\mathrm{Var}(E(\theta|\text{chip data}))$ is estimated by the observed variance in the imputed expected counts and $\mathrm{Var}(\theta)$ was estimated by $p(1-p)$, where $p$ is the allele frequency.

**Gene and variant annotation.** Variants were annotated using Ensembl release 80 and Variant Effect Predictor version 2.8[38]. A total of 32.5 million variants passed the quality threshold and were imputed into 151,677 Icelanders who had been genotyped using Illumina chips.

**Adjusting for relatedness**. To account for inflation in test statistics due to cryptic relatedness and stratification, we applied the method of LD score regression[39]. With a set of 1.1 M variants, we regressed the $\chi^2$ statistics from our GWA scan against LD score and used the intercept as a correction factor. The LD scores were downloaded from an LD score database (ftp://atguftp.mgh.harvard.edu/brendan/1k_eur_r2_hm3snps_se_weights.RDS; accessed 23 June 2015).

**Thresholds for genome-wide significance**. The threshold for genome-wide significance was corrected for multiple testing with a weighted Bonferroni adjustment using as weights the enrichment of variant classes with predicted functional impact among association signals[14]. With 32,463,443 sequence variants being tested, the weights given in Sveinbjornsson et al.[14] were rescaled to control the family-wise error rate. This yielded significance thresholds of $2.6 \times 10^{-7}$ for high-impact variants ($N = 8,464$), $5.1 \times 10^{-8}$ for moderate-impact variants ($N = 149,983$), $4.6 \times 10^{-9}$ for low-impact variants ($N = 2,283,889$), $2.3 \times 10^{-9}$ for other variants in Dnase I hypersensitivity sites ($N = 3,913,058$) and $7.9 \times 10^{-10}$ for other variants ($N = 26,108,039$).

**Association analysis**. Logistic or linear regression were used to test for the association of SNPs with binary or quantitative traits, respectively, treating disease status or quantitative trait as the response and allele counts from direct genotyping or expected genotype counts from imputation as covariates. To account for inflation in test statistics due to cryptic relatedness and stratification, we applied the method of LD score regression[39]. The estimated correction factor for AS based on LD score regression was 1.19 for the additive model.

**Genetic risk score**. Weighted GRS for CAD (CAD-GRS-*all*) was generated based on previously reported CAD effect estimates (logOR) (see Supplementary Data 4). Variants used for the CAD-GRS-*lip* included a subset of variants in CAD-GRS that associated with non-HDL cholesterol ($P < 1 \times 10^{-8}$) in the Icelandic dataset, or are located at the *LPA* locus. The two *LPA* locus variants included in CAD-GRS-*lip* associate with lipoprotein(a) at $P < 8 \times 10^{-90}$ in Iceland.

**Genetic association replication studies**. The Malmo Diet and Cancer Study (MDCS) is a community-based prospective cohort of middle-aged individuals from Southern Sweden. In total, 30,447 subjects attended a baseline exam in 1991–1996 when they filled out a questionnaire, underwent anthropometric measurements, and donated peripheral venous blood samples[40]. Prevalent or incident cases of AS were ascertained from nationwide hospital registers with high validity as described previously[19]. Genome-wide genotyping of single-nucleotide variants was performed using the Illumina Human Omni Express Exome BeadChip kit. Genotyping was performed in a nested case-cohort design including 15,362 subjects with complete data, of which 470 cases with incident AS. The SNP rs10455872 was genotyped in the entire cohort, with genotypes available in 28,722 subjects, including 613 cases with incident AS. Association with incident AS was tested in a case–control analysis utilizing logistic regression under an additive inheritance model adjusted for age and sex. Case–control matching was performed in SAS v9.4 with the greedy algorithm, matching 1 AS case to 1 population-based controls for sex, baseline age (<3 years age difference), year of baseline visit (within 3 years from visit), and requiring at least equal follow-up in controls. All participants were of European ancestry, confirmed by multidimensional scaling of genome-wide data. Informed consent was obtained from all participants and the study was approved by the Ethics Committee of Lund University, Sweden.

Patients from the greater Stockholm area with AS, BAV, or TAA were recruited as a part of the ASAP (the Advanced Study of Aortic Pathology) and Artist studies. The ASAP cohort consists of 429 patients undergoing aortic valve surgery at the Karolinska University Hospital (Stockholm, Sweden)[29,41]. The samples were genotyped on Illumina 610wQuad beadchips and approximately 588,400 SNPs were provided after quality control (QC). The Artist cohort consists of 406 samples genotyped with Omni-2.5 Quad beadchips on 2,443,180 SNPs. Imputation was performed using Impute2 from 1000G phase1 v3. Samples from the POLCA/Olivia cohorts were used as controls (a total of 1,295 individuals). POLCA consists of healthy 50-year-old men, free from coronary heart disease, recruited at random using the population registry. The POLCA samples were genotyped on Illumina 610kwQuad. The Olivia comprises both men and women with an age distribution 33–80 years. In Olivia, 670 control samples were genotyped on illumina 1M genotyping arrays. The vast majority of included control samples are of Scandinavian ancestry. Association analysis was performed using SNPTEST, with age, sex, and first 10 principal components as covariates. The study was approved by the Human Research Ethics Committee at Karolinska Institutet (ASAP study, ethical approval number 2006/784-31/1; Artist cohort, 2008/1771-31; POLCA/Olivia control samples, 03-491), Stockholm, Sweden.

The Norwegian Nord-Trøndelag Health Study (HUNT) is a population-based health survey conducted in the county of Nord-Trøndelag, Norway. Individuals were included at three different time points during approximately 20 years (HUNT1 (1984–1986), HUNT2 (1995–1997), and HUNT3 (2006–2008))[42]. At each time point, the entire adult population (≥20 years) was invited to participate by completing questionnaires, attending clinical examinations, and interviews. Taken together, the health studies include information from over 120,000 different

individuals from Nord-Trøndelag. Biological samples including DNA have been collected for approximately 70,000 participants. AS was defined based on ICD-10 codes collected from local hospitals and out-patient clinics between 1999 and 2016. Cases were defined as individual with one or more ICD-10 codes specific for AS ("I35.0" or "I35.2"), whereas controls were all individuals without a code specific for AS. In total, 1,546 cases with AS and 24,235 controls genotyped with Illumina HumanCoreExome arrays were analyzed. Association analysis were conducted using EPACTS-3.3. The SNP-phenotype associations were modeled using the Firth Bias-Corrected Logistic Likelihood Ratio Test[43], assuming an additive genetic model for genotyped markers and imputed genotypes. Models were adjusted for sex, birth year, genotyping batch, and four principal components (PCs). PCs were computed using PLINK. Individuals of non-European ancestry, based on principal components analysis (PCA) were excluded from the study. Additional filters applied to the analysis included minor allele count ≥10 and imputation $r^2 \geq 0.3$. Participation in the HUNT Study is based on informed consent, and the study has been approved by the Data Inspectorate and the Regional Ethics Committee for Medical Research in Norway.

In the years 2006–2010, the UK Biobank study recruited 502,647 individuals aged 37–76 years from across the country. All participants provided information regarding their health and lifestyle via touch screen questionnaires, consented to physical measurements, and agreed to have their health followed. They also provided blood, urine, and saliva samples for future analysis. UK Biobank has ethical approval from the Northwest Multi-Center Research Ethics Committee, and informed consent was obtained from all participants. Genotype imputation data was available for 487,409 individuals (May 2017 release), of which 408,658 were used in the analysis. The 408,658 individuals were selected as self-reported white British with similar genetic ancestry based on principal component analysis and with consistent reported and genetically determined gender. AS was defined according to ICD-10 codes I35.0 and I35.2, based on diagnoses codes a participant has had recorded across all their episodes in hospital, and CAD was defined as the codes I20.0, I21, I22, I25.0, I25.1, I25.2, and I25.9. The case–controls analysis was done using SNPTEST v2.5.2[44] and the association with the GRS was tested using R v3.4.1[45]. In both cases, the analysis was adjusted for age, gender, and 20 principle components. To adjust for relatedness and remaining population stratification the $P$ values were adjusted using genomic control adjustment, with adjustment factors $\lambda_g$ estimated based on association analysis of 155,000 unrelated common variants. The estimated $\lambda_g$'s were 1.023, 1.088, 1.029, and 1.013 for the analysis of AS, CAD, AS restricted to CAD and AS excluding CAD, respectively.

In the University of Michigan and Cardiovascular Health Improvement Study, we collected DNA from consented individuals with BAV from the Frankel Cardiovascular Center at the University of Michigan as part of the University of Michigan BAV registry or the Cardiovascular Health Improvement Project (CHIP). Patients were typically seen in clinic for aortic valve replacement or aortic aneurysm. DNA was isolated from peripheral blood lymphocytes. Four hundred and seventy-three BAV cases, 251 AS cases, and 809 TAA cases were successfully collected and genotyped. We identified potential controls from a surgical-based biobank, the Michigan Genomics Initiative (MGI), that were genotyped with the same GWAS array as cases. After excluding those with aortic disease, we performed age matching by requiring controls to have a birth year within −5 and +10 years of the case. From the available controls in the appropriate age and sex category, we selected the best ethnic match for each case and repeated the greedy algorithm until a control was selected for each case. We repeated the entire process so that 10 controls were selected for each AS and BAV case and 5 controls for each TAA case. All MGI research subjects provided informed consent. We performed genotyping using a GWAS+exome chip array (Illumina HumanCoreExome). To avoid any potential batch effects, cases and controls were genotyped using the same array in the same genotyping center (Sequencing and Genotyping core at the University of Michigan). Genotype calling was performed using GenTrain version 2.0 in GenomeStudio V2011.1 (Illumina) using identical cluster files for cases and controls. Samples with <98% genotype calls, evidence of gender discrepancy, and duplicates as well as individuals with non-European ancestry identified by plotting the first 10 genotype-driven principal components were excluded from further analysis. We performed variant-level QC by excluding variants that met any of the following criteria; variants with a cluster separation score <0.3, <98% genotype call rate, or deviation from Hardy–Weinberg equilibrium ($P < 1 \times 10^{-5}$). We phased the autosomal genotype data using SHAPEIT2[46] and imputed variants from the Haplotype Reference Panel v1[47] using minimac3[48,49]. We excluded poorly imputed variants with imputation R2 < 0.3. We performed single-variant association testing for BAV, TAA, and AS status using the Wald test based on logistic regression with age, sex, and the first four principal components as covariates using the EPACTS software (URL: http://csg.sph.umich.edu/kang/epacts/) for imputed dosages. All repository projects utilized for this study are approved by the University of Michigan, Medical School, Institutional Review Board, and informed consent was obtained from study participants.

In the BAVCon consortium, 452 sporadic self-reported Caucasian BAV cases were genotyped using the Illumina Omni-2.5 platform. One thousand eight hundred and thirty-four self-reported Caucasian population controls from the Framingham Heart Study genotyped using the Omni5 platform. Caucasian ancestry was further identified using PCA to detect clusters, filter outliers, and filter-related individuals both before and after merging cases and controls for association analysis. Principal components were included as covariates in

association analysis. Further QC of the genotype data from both cohorts was performed using GenomeStudio and PLINK. After QC, we imputed additional genotypes against the 1,000 genomes reference (Phase 3) using IMPUTE2 to yield 7,913,553 genetic markers for an additive logistic regression model, adjusted for gender and age. This study has been approved by Partner's in HealthCare Human Research Committee, and informed consent was obtained from study participants.

In the University of Texas Health Science Center, 765 patients of European descent with TAA or aortic dissections, were enrolled and genotyped[50]. A subset of these patients had BAV, N = 147. The study included 864 controls from US National Institute of Neurological Disorders and Stroke (NINDS)[51]. We imputed additional genotypes against 1,000 genomes reference (Phase 3), and for association analysis, we used additive logistic regression model accounting for gender and principal components[51]. Subjects of non-European ancestry according to multidimensional scaling were removed from the analysis. This study has been approved by Committee for the Protection of Human Subjects at UT Health Science Center at Houston, and informed consent was obtained from study participants.

**Identification of candidate causal variants and genes**. Epigenome data from the NIH Roadmap Epigenome Mapping Consortium (http://egg2.wustl.edu/roadmap/data/byFileType/chromhmmSegmentations/ChmmModels/imputed12marks/jointModel/final/catMat/hg19_chromHMM_imputed25.gz) for 11 histone marks analyzed by chromatin immunoprecipitation with sequencing, together with open chromatin regions analyzed by DNase-seq, integrated into 25 discrete chromatin states through ChromHMM were downloaded[26]. Lead variants and those found in LD ($R^2 > 0.5$) within the AS loci were then annotated for chromatin states involving regulatory functions, that is, EnhA (Active enhancer elements), EnhW (Weak enhancer elements), EnhTx (Enhancer marks coupled with transcription-associated histone marks), DNAse (Open chromatin configuration), TssA (Active transcription start site), PromUp/D (Regions upstream or downstream of promoter), PromP (Poised promoter), PromBiv (Bivalent promoters), Het (Heterochromatin), ReprPC (Polycomb-group repressed) while omitting states indicative of transcription status only (Tx3′, Tx5′, Tx, TxWk) or states characteristic of zinc-finger protein genes (ZNF/Rpts).

High-throughput 3C analysis (Hi-C) data for right ventricular and left ventricular tissue and aorta were obtained from an online repository: Functional Mapping and Annotation of Genome-Wide Association Studies (FUMA GWAS: http://fuma.ctglab.nl/)[52]. It uses public datasets compiled by Schmitt et al.[28] in order to identify structural interactions of enhancers with genes (Hi-C compendium: https://www.ncbi.nlm.nih.gov/geo/query/acc.cgi?acc = GSE87112)

Gorpipe[53] was employed to query the data and then imported into R for further analysis and plotting making use of base functions (R 3.4.1). The Hi-C data are binned into intervals of 40 kb which are indicated as gray horizontal lines in lower panels of Fig. 1a, b, and their midpoints were then used to draw arcs from one midpoint to another for interactions at false-discovery rate $<1 \times 10^{-10}$. Intersection between chromatin interaction intervals and promoters was based on Refseq annotated transcription start sites.

**Data availability**. The Icelandic population WGS data has been deposited at the European Variant Archive under accession code PRJEB8636. The authors declare that the data supporting the findings of this study are available within the article, its Supplementary Data files and upon request.

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

## Acknowledgements

We are grateful to all study subjects for their valuable participation in our research. We thank our collaborators and the staff at deCODE genetics core facilities and the Research Service Center for their important contribution to this work. We would like to thank all UK Biobank participants, staff, and investigators. This research has been conducted using the UK Biobank Resource under Application Number 15293. We thank all participants of the Cardiovascular Health Improvement Project (CHIP) and BAV registry at the University of Michigan for their contribution to research. We appreciate the valuable efforts from the CHIP and BAV registry collection team. The University of Michigan Health System—Cardiovascular Health Improvement Project was supported by the Frankel Cardiovascular Center. We appreciate the Aikens Fund for Aortic Research (to C.J.W., B.Y.) and McKay research award for supporting this project (to B.Y.). B.Y. is supported by American Association of Thoracic Surgery (AATS) Graham Foundation and Thoracic Surgery Foundation of Research and Education (TSFRE). C.J.W. is supported by HL109946, HL130705, and HL127564. W.Z. is supported by the Rackham Predoctoral Fellowship of the University of Michigan.

## Author contributions

A.H., H.H., D.G., G.Thl., U.Th., and K.S. designed the study and interpreted the results. A.H., H.H., S.G., and G.Thl. drafted the manuscript. A.H., G.Thl., D.G. O.A.S., and V.T. performed bioinformatic and statistical analysis in the discovery study and joint data analysis. H.H., I.J., R.B.Th., Th.B., V.S., G.O., I.O., E.L.S., R.D., T.G., A.G., and G.Th. collected and characterized Icelandic phenotype data. P.A., O.M., J.G.S., P.E, A.Ha, A.F.-C., L.Fr., K.H, C.N.-C., D.M., D.G., B.Y., W.H., C.J.W., M.L., C.M.B, G.A., M.M., and S.C.B. designed and managed individual studies and characterized phenotype data. A.M., L. Fo., J.B.N., N.V., S.P., W.Z., and M.H., performed statistical analyses for individual replication studies. A.H., G.Thl., H.H., D.G., G.Thl, U.Th., K.S., S.G., O.A.S., V.T., I.J., R. B.Th., Th.B., V.S., G.O., I.O., E.L.S., R.D., T.G., A.G., P.A., O.M., J.G.S., P.E, A.Ha, A.F.-C., L.Fr., K.H, C.N.-C., D.M., D.G., B.Y., W.H., C.J.W., M.L., C.M.B, G.A., M.M., S.C.B., A.M., L.Fo., J.B.N., N.V., S.P., W.Z., and M.H. reviewed the manuscript. K.S. supervised the study.

## Additional information

**Competing interests:** A.H., S.G., G.Thl, O.A.S., V.T., R.B.Th, I.J., Th.B., V.S., G.Th., U. Th., D.G., H.H., and K.S. are employed by deCODE Genetics/Amgen Inc. All the other authors declare no competing financial interests.

Anna Helgadottir [1], Gudmar Thorleifsson [1], Solveig Gretarsdottir [1], Olafur A. Stefansson [1], Vinicius Tragante [1], Rosa B. Thorolfsdottir [1], Ingileif Jonsdottir [1,2], Thorsteinn Bjornsson [1], Valgerdur Steinthorsdottir [1], Niek Verweij [3,4], Jonas B. Nielsen [5], Wei Zhou [6], Lasse Folkersen [7,8], Andreas Martinsson [9], Mahyar Heydarpour [10], Siddharth Prakash [11], Gylfi Oskarsson [12], Tomas Gudbjartsson [13], Arnar Geirsson [14], Isleifur Olafsson [15], Emil L. Sigurdsson [16,17], Peter Almgren [18,19], Olle Melander [18,19], Anders Franco-Cereceda [20], Anders Hamsten [7], Lars Fritsche [21,22], Maoxuan Lin [5], Bo Yang [23,24], Whitney Hornsby [24], Dongchuan Guo [11], Chad M. Brummett [25], Gonçalo Abecasis [26], Michael Mathis [25], Dianna Milewicz [11,27], Simon C. Body [10], Per Eriksson [7], Cristen J. Willer [5,6,24,28], Kristian Hveem [21,22], Christopher Newton-Cheh [4,29,30], J. Gustav Smith [9], Ragnar Danielsen [2,31], Gudmundur Thorgeirsson [1,2,31], Unnur Thorsteinsdottir [1,2], Daniel F. Gudbjartsson [1,32], Hilma Holm [1] & Kari Stefansson [1,2]

[1]deCODE genetics/Amgen Inc., Reykjavik, 101 Iceland. [2]Faculty of Medicine, University of Iceland, Reykjavik, 101 Iceland. [3]Department of Cardiology, University of Groningen, University Medical Center Groningen, 9700 RB Groningen, The Netherlands. [4]Medical and Population Genetics Program, Broad Institute of MIT and Harvard, Cambridge, 02142 MA, USA. [5]Department of Internal Medicine, Division of Cardiovascular Medicine, University of Michigan, Ann Arbor, 48109 MI, USA. [6]Department of Computational Medicine and Bioinformatics, University of Michigan,

Ann Arbor, 48109 MI, USA. [7]Cardiovascular Medicine Unit, Department of Medicine, Karolinska University Hospital Solna, Karolinska Institutet, Stockholm, 17176 Sweden. [8]Department of Bioinformatics, Technical University of Denmark, Copenhagen, 2800 Denmark. [9]Department of Cardiology, Clinical Sciences, Lund University and Skåne University Hospital, Lund, 22185 Sweden. [10]Department of Anesthesiology, Perioperative and Pain Medicine, Brigham and Women's Hospital, 75 Francis Street, Boston, MA 02115 USA. [11]Department of Internal Medicine, Division of Medical Genetics, University of Texas Health Science Center at Houston, Houston, 77030 TX, USA. [12]Childrens Hospital, Landspitali National University Hospital of Iceland, Reykjavik, 101 Iceland. [13]Department of Surgery and Cardiothoracic Surgery, Landspitali National University Hospital, Reykjavik, 101 Iceland. [14]Section of Cardiac Surgery, Department of Surgery, Yale University School of Medicine, New Haven, 06510 CT, USA. [15]Department of Clinical Biochemistry, Landspitali National University Hospital, Reykjavik, 101 Iceland. [16]Heilsugaeslan Solvangi, Hafnarfjördur, 220 Iceland. [17]Department of Family Medicine, University of Iceland, Reykjavik, 101 Iceland. [18]Department of Clinical Sciences, Lund University, Malmö, 22185 Sweden. [19]Department of Internal Medicine, Skåne University Hospital, Malmö, 22185 Sweden. [20]Cardiothoracic Surgery Unit, Department of Molecular Medicine and Surgery, Karolinska University Hospital Solna, Karolinska Institutet, Stockholm, 17176 Sweden. [21]HUNT Research Centre, Department of Public Health and General Practice, Norwegian University of Science and Technology, Levanger, 7491 Norway. [22]K.G. Jebsen Center for Genetic Epidemiology, Department of Public Health, Norwegian University of Science and Technology, Trondheim, 7491 Norway. [23]Department of Cardiac Surgery, University of Michigan, Ann Arbor, MI 48105, USA. [24]Frankel Cardiovascular Center, University of Michigan, Ann Arbor, MI 48109, USA. [25]Department of Anesthesiology, University of Michigan, Ann Arbor, MI 48105, USA. [26]Department of Biostatistics, University of Michigan, Ann Arbor, MI 48109, USA. [27]Medicine Services, Texas Heart Institute, St. Luke's Episcopal Hospital, Houston, TX 77030, USA. [28]Department of Human Genetics, University of Michigan, Ann Arbor, MI 48109, USA. [29]Massachusetts General Hospital, Harvard Medical School, Broad Institute of Harvard and MIT, Boston, MA 02114, USA. [30]Cardiovascular Research Center, Massachusetts General Hospital, Boston, MA 02114, USA. [31]Department of Internal Medicine, Division of Cardiology, Landspitali National University Hospital of Iceland, Reykjavik, 101 Iceland. [32]School of Engineering and Natural Sciences, University of Iceland, Reykjavik, 101 Iceland

