## [Peer Review File · Nature Communications]

Reviewers' comments:

Reviewer #1 (Remarks to the Author):

This is a very well executed study with a large discovery cohort and multiple replication populations that reports three novel loci for aortic valve stenosis, and confirms previous associations in LPA and MYH6. The methods are well described and findings appropriately presented in context of existing knowledge. The finding of variants that associate with congenital heart defects and AVS helps confirm thinking in that field that these are related. Similarly this work also shows strong association with atherosclerotic disease and AVS. It is interesting there was no association with several well known genes that cause AVS and are associated with BAV and calcific valve disease, such as NOTCH1, and the osteogenic pathway (scleraxis, RUNX2 etc.). This would merit mention in the paper.

It would be helpful to include Manhattan plots of their initial Icelandic cohort.

Could the authors state how the 6 variants in the Icelandic cohort were chosen for replication?

Bioinformatics analysis of epigenetic, expression and regulatory data help provide support for the associated variants. The paper could be strengthened by the addition of functional data showing the associated variants, or variants in LD or in the same TAD, had demonstrated effect.

Methods: The authors do not describe populations used for the replication cohorts. Were these confined to just individuals of European ancestry? If not, could the authors describe what was done to control for population stratification?

Supplementary tables: Please include confidence intervals for all odds ratios reported

Reviewer #2 (Remarks to the Author):

The authors present a GWAS of aortic stenosis (AS) in a new, large combined sample of 7,307 cases and 801,073 controls. They report associations of two novel loci with AS risk, and confirmation of two previously associated loci. Using refreshingly uncomplicated methods, they explore the contributions of CAD-associated variants to AS risk, taking into account the influences of lipid-associated loci. Their conclusions that the aetiology of AS is driven at least in part by atherosclerosis-related and foetal development pathways is well justified by the analysis.

In general, the analysis is well described and executed and the findings are clearly and concisely explained. The authors make appropriate inferences from the results of their analysis.

I have a small number of comments on the manuscript. (Page and line numbers refer to those marked in the PDF file).

Page 3 l86 - "We tested rs7543130 and six additional variants...": The authors should clarify why these additional six variants were selected for validation. The cited tables in their current form do not appear to provide sufficient explanation.

Page 4 l100-1 - "The effect sizes...": The authors should note here that although they were directionally consistent, the significance of the associations was weaker in the surgical AVR sample, albeit probably as a result of lower case numbers.

Page 4 l102 - "...studies show that...": The authors should be more specific about the nature of these studies. e.g. traditional histology, molecular phenotyping.

Page 4 l107 - "A rare p.Arg721Trp MYH6...": To aid the reader, the rsID corresponding to the MYH6 variant should be included in the text as well as in the tables.

Page 4 l108-9 – "...previously shown to associate with cardiac arrhythmias...": The authors should be more specific about the nature of the associated cardiac arrhythmias. Whether these are atrial, ventricular or other arrhythmias would aid the reader's interpretation.

Given the authors' argument that a large proportion of the aetiology of AS operates through Lp(a) or non-HDL-cholesterol, it would be helpful to present updated Mendelian randomisation analyses in this new, larger dataset that seek to refine the causal estimates for those those two lipid fractions.

Reviewer #3 (Remarks to the Author):

This is the first large GWAS for aortic stenosis including sizable discovery and replication cohorts.

The findings of interest include replication of the LPA locus and an MYH6 coding variant and identification of two novel loci near PALMD and intronic to TEX41. The same variants are found to associate with BAV, aortic root size, ventricular/atrial septal defects.

The authors also nicely demonstrate that genetic variants related to the risk of both CAD and AS are confined to those affecting Lp(a) and LDL-C concentrations consistent with the known effect of these lipids on AS risk.

The relationship with BAV is notably stronger than with AS and may drive the latter association. For any of the different AS cohorts, are individual data on the coexistence of BAV available?

In the absence of BAV, does the novel PALMD locus associate with aortic stenosis?

With the exception of loci associated with Lp(a) or LDL-C levels, it is not clear that the GWAS findings are for aortic stenosis per se rather than BAV, aortic root size, ventricular/atrial septal defects. This needs to be clarified.

The analysis and discussion on possible functional effects is rather cursory and functional studies are lacking. There are no experimental data, rather interrogation of available database. In any case, the authors could add that region in tight LD with rs1830321 includes SNPs with multiple regulatory signals (Haploregv4). Recent GTEx data indicate a significant association of rs18303221 with TEX41 expression (albeit currently limited to thyroid). This is consistent with findings from chromatin interaction mapping. It would also be useful to request a look-up from other groups have eQTL data in left ventricle (e.g.PMID: 24846176).

Response to reviewers' comments:

We thank the reviewers for constructive comments and suggestions. We have revised the manuscript based on these suggestions and you will find our point-by-point response below.

We note that we have re-formatted the manuscript to accommodate Nature Communications manuscript checklist for article format.

Point-by-point responses:

Reviewer #1 (Remarks to the Author):

This is a very well executed study with a large discovery cohort and multiple replication populations that reports three novel loci for aortic valve stenosis, and confirms previous associations in LPA and MYH6. The methods are well described and findings appropriately presented in context of existing knowledge. The finding of variants that associate with congenital heart defects and AVS helps confirm thinking in that field that these are related. Similarly this work also shows strong association with atherosclerotic disease and AVS.

It is interesting there was no association with several well know genes that cause AVS and are associated with BAV and calcific valve disease, such as NOTCH1, and the osteogenic pathway (scleraxis, RUNX2 etc.). This would merit mention in the paper.

Response: To address this comment we have added the following sentence to Results, under the subheading Novel variants associate with aortic stenosis: *"In contrast, we did not find association with variants implicating osteogenic and calcium-signaling pathway genes, previously reported to associate suggestively with AS¹⁶ (P>0.05 in Iceland and UK Biobank)".*

It would be helpful to include Manhattan plots of their initial Icelandic cohort.

Response: The manuscript now includes a Manhattan plot of the discovery aortic valve stenosis analysis and we refer to this figure in Results: *"We tested 32.5 million sequence variants for association with AS in 2,457 Icelandic cases and 349,342 controls (see Manhattan plot in Supplementary Fig. 1)."*

Could the authors state how the 6 variants in the Icelandic cohort were chosen for replication?

Response: The rs7543130 and six additional variants were selected for replication as they represented the top seven common or low frequency variants with the most significant association in the genome wide scan in Iceland. This is now clearly stated in the manuscript (Results section).

“We tested the top seven common and low frequency variants in the discovery GWA scan, including rs7543130...”

Bioinformatics analysis of epigenetic, expression and regulatory data help provide support for the associated variants. The paper could be strengthened by the addition of functional data showing the associated variants, or variants in LD or in the same TAD, had demonstrated effect.

Response: We agree with the reviewer that functional data showing how the associated variants affect the pathogenesis of disease is of importance and should be pursued. However, since the suggested work is a major undertaking, and mostly out of our area of expertise, we feel it is better addressed in separate studies.

Methods: The authors do not describe populations used for the replication cohorts. Were these confined to just individuals of European ancestry? If not, could the authors describe what was done to control for population stratification?

Response: We thank the reviewer for pointing out that this was not clear enough in our previous version of the manuscript. In our revised version, the ancestry of the study populations are more clearly described in the Methods section (see below), and specifically stated in the Abstract.

Added to Abstract: *“Here we report a large genome-wide association (GWAS) study of 2,457 Icelandic AS cases and 349,342 controls with a follow-up in up to 4,850 cases and 451,731 controls of European ancestry.”*

Added to Methods:

- a. Sweden-The Malmo Diet and Cancer Study (MDCS): AS study

“All participants were of European ancestry, confirmed by multidimensional scaling of genome-wide data.”

- b. Norway - The Norwegian Nord-Trøndelag Health Study (HUNT): AS study

“Individuals of non-European ancestry, based on PCA were excluded from the study.”

- c. United States – Houston (STAAD study): BAV and TAA study

“Briefly, we used additive logistic regression model accounting for gender and principal components. Subjects of non-European ancestry according to multidimensional scaling were removed from the analysis.”

- d. United States – Boston (BAVCon consortium): BAV study

The language has been revised to clarify that only individuals of European ancestry were included and that principal components were included as covariates in the association analysis.

Supplementary tables: Please include confidence intervals for all odds ratios reported

Response: Confidence intervals for the odds ratios are now reported in all Supplementary tables.

Reviewer #2 (Remarks to the Author):

The authors present a GWAS of aortic stenosis (AS) in a new, large combined sample of 7,307 cases and 801,073 controls. They report associations of two novel loci with AS risk, and confirmation of two previously associated loci. Using refreshingly uncomplicated methods, they explore the contributions of CAD-associated variants to AS risk, taking into account the influences of lipid-associated loci. Their conclusions that the aetiology of AS is driven at least in part by atherosclerosis-related and foetal development pathways is well justified by the analysis. In general, the analysis is well described and executed and the findings are clearly and concisely explained. The authors make appropriate inferences from the results of their analysis.

I have a small number of comments on the manuscript. (Page and line numbers refer to those marked in the PDF file).

Page 3 l86 – “We tested rs7543130 and six additional variants...”: The authors should clarify why these additional six variants were selected for validation. The cited tables in their current form do not appear to provide sufficient explanation.

Response: This is now clearly stated in the manuscript (Results section). *“We tested the top seven common and low frequency variants in the discovery GWA scan, including rs7543130...”*

Page 4 l100-1 - “The effect sizes...”: The authors should note here that although they were directionally consistent, the significance of the associations was weaker in the surgical AVR sample, albeit probably as a result of lower case numbers.

Response: To clarify this we have modified the following sentence:

“We tested the association of the two novel AS variants and the LPA variant, with a subset of Icelandic AS cases who had undergone aortic valve replacement, representing those with severe AS. Although less significant, likely due to smaller sample size, the effects were not significantly different from those for all AS (Supplementary Table 2).”

Page 4 l102 – “...studies show that...”: The authors should be more specific about the nature of these studies. e.g. traditional histology, molecular phenotyping.

Response: The sentence now reads:

“However, several of the associated clinical risk factors of calcific AS are shared with atherosclerotic disease, and immunohistochemical studies show that calcified aortic valve lesions have many characteristic features of atherosclerosis,...”

Page 4 l107 – “A rare p.Arg721Trp MYH6...”: To aid the reader, the rsID corresponding to the MYH6 variant should be included in the text as well as in the tables.

Response: The rs-ID has been added to the text and is now displayed more prominently in Table 2.

Page 4 l108-9 – “...previously shown to associate with cardiac arrhythmias...”: The authors should be more specific about the nature of the associated cardiac arrhythmias. Whether these are atrial, ventricular or other arrhythmias would aid the reader's interpretation.

Response: We have made this clearer in the sentence, which now reads: *“...previously shown to associate with sick sinus syndrome and atrial fibrillation...”*

Given the authors’ argument that a large proportion of the aetiology of AS operates through Lp(a) or non-HDL-cholesterol, it would be helpful to present updated Mendelian randomisation analyses in this new, larger dataset that seek to refine the causal estimates for those two lipid fractions.

Response: We agree with the reviewer and acknowledge the clinical importance of estimating the effects of these causal risk factors on AS. However we feel that this should be pursued in separate studies to allow for enough room for adequate description and discussion of analyses and results.

To this end, we (deCODE authors) have recently submitted a manuscript with detailed Mendelian randomization (MR) analyses of Lp(a) for several cardiovascular phenotypes, with focus on coronary artery disease, but including AS. Our MR analyses using five LPA variants as instruments confirm the causal relationship between Lp(a) and AS and indicate that the effect of sequence variants at LPA on AS risk is proportional to their effect on Lp(a) molar concentration. We found that Lp(a) molar concentration associates in a dose-dependent manner with AS risk with 17% increase in risk per 50 nM Lp(a) increase (95%CI:12-22%).

We have examined non-HDL cholesterol genetic risk score (grs) for association with AS in the Icelandic data. The non-HDL cholesterol grs was based on 27 rare and low-frequency blood lipid variants combined with 185 known lipid-associated variants (as described in Nat.Genet.2016, 48(6):634-9. *Variants with large effects on blood lipids and the role of cholesterol and triglycerides in coronary disease*).

The results indicate that the risk of AS is increased by 41% (P=6.6e-09), per 1 standard deviation increase in non-HDL cholesterol. In line with the results shown for CAD-GRS-lip in the current manuscript, the non-HDL cholesterol grs increases the risk of AS after adjusting for CAD diagnoses (P=7.3e-05).

Reviewer #3 (Remarks to the Author):

This is the first large GWAS for aortic stenosis including sizable discovery and replication cohorts. The findings of interest include replication of the LPA locus and an MYH6 coding variant and identification of two novel loci near PALMD and intronic to TEX41. The same variants are found to associate with BAV, aortic root size, ventricular/atrial septal defects.

The authors also nicely demonstrate that genetic variants related to the risk of both CAD and AS are confined to those affecting Lp(a) and LDL-C concentrations consistent with the known effect of these lipids on AS risk.

The relationship with BAV is notably stronger than with AS and may drive the latter association.

For any of the different AS cohorts, are individual data on the coexistence of BAV available?

In the absence of BAV, does the novel PALMD locus associate with aortic stenosis?

Response: Unfortunately, we have very limited information on the coexistence of BAV among the AS cases such that testing variants for association with AS in the absence of BAV is not possible in our samples.

The possibility that BAV drives the association of AS, and the limitation of our study to specifically test this, is now discussed in our re-formatted manuscript (Discussion):

“Given that BAV is a major risk factor for AS, and that the MYH6 and chromosome 1p21 variants have substantially greater effects on BAV than on AS, it may be postulated that the AS risk conferred by these variants is mediated through BAV. However, as we had limited information on whether AS occurred on the background of bicuspid or tricuspid valve, we were not able to examine whether these variants associate with AS in absence of BAV..”

With the exception of loci associated with Lp(a) or LDL-C levels, it is not clear that the GWAS findings are for aortic stenosis per se rather than BAV, aortic root size, ventricular/atrial septal defects. This needs to be clarified.

Response: See response to the comment above regarding the possibility that BAV drives the association.

Attempting to clarify this issue, we now discuss our interpretation of the data with respect to these phenotypes in our re-formatted manuscript (Discussion):

“Interestingly, the AS and BAV risk allele of rs7543130 near PALMD also associates with increased aortic root size. The relationship between BAV and aortopathy is well recognized and several studies suggest that the dilation of the proximal ascending aorta results from changes in flow conditions secondary to the presence of BAV^{5,23,33}. This raises the question whether the effect of chromosome 1p21 variant on aortic root size can be explained by its association with BAV. However, our results indicate that the variant’s impact on aortic root size is not merely a consequence of BAV, since excluding BAV cases from the analysis had minimal effect on the association. We also note that the MYH6 missense variant has a large effect on BAV but no effect on aortic root size. While we did not

observe a consistent genetic association between risk of AS and aortic root size we identified one additional aortic root size variant, rs17696696 intronic to CFDP1 that associated with AS.”

The analysis and discussion on possible functional effects is rather cursory and functional studies are lacking.

There are no experimental data, rather interrogation of available database. In any case, the authors could add that region in tight LD with rs1830321 includes SNPs with multiple regulatory signals (Haploregv4). Recent GTEx data indicate a significant association of rs18303221 with TEX41 expression (albeit currently limited to thyroid). This is consistent with findings from chromatin interaction mapping. It would also be useful to request a look-up from other groups have eQTL data in left ventricle (e.g.PMID: 24846176).

Response: In response to the first part of this comment, we illustrate this point in Figure 1 where variants in LD with the lead association variant are found to overlap with regulatory regions as defined by Chrom-HMM 25-state model. The data underlying the 25-state model, heart and aorta tissue types, are derived from Roadmap Epigenomics data and therefore the same as in Haploreg. In response to the second point, we have looked at the data from the publication mentioned here (PMID: 24846176), which is available through GEO omnibus. Consistent with results from GTEx that we report in our study, the associated variants, or variants in LD with the lead variant, do not overlap with eQTL signals in this study (PMID: 24846176).

We modified the description of results from GTEx in the manuscript, given the most recent GTEx data:

“Assessment of 44 diverse human tissues from adults indicated a significant association of rs1830321 with TEX41 expression, albeit limited to thyroid tissue, but no eQTLs were observed for rs7543130.”

REVIEWERS' COMMENTS:

Reviewer #1 (Remarks to the Author):

My previous critiques have been adequately addressed. I have no new comments.

Kim McBride

Reviewer #2 (Remarks to the Author):

The authors have responded comprehensively and adequately to my comments and implemented suggested changes appropriately. I take their point about their other submitted manuscript concerning Mendelian randomisation and this should be a sufficiently clear way of presenting those findings alongside the present manuscript; it would be helpful if the two published papers could reference each other, if possible.

I have no further comments.

Point-by-point responses to reviewers' comments:

Reviewer #1 (Remarks to the Author):

My previous critiques have been adequately addressed. I have no new comments.

Kim McBride

Reviewer #2 (Remarks to the Author):

The authors have responded comprehensively and adequately to my comments and implemented suggested changes appropriately. I take their point about their other submitted manuscript concerning Mendelian randomisation and this should be a sufficiently clear way of presenting those findings alongside the present manuscript; it would be helpful if the two published papers could reference each other, if possible.

I have no further comments.

Response:

We thank the reviewer for this reasonable suggestion. We aim to refer to the current manuscript in our submitted lipoprotein(a) Mendelian randomization study. However, as our recent lipoprotein(a) study has not been accepted for publication yet, it is difficult for us to refer to it in our current manuscript.